# Flagellin Improves the Immune Response of an Infectious Bursal Disease Virus (IBDV) Subunit Vaccine

**DOI:** 10.3390/vaccines10111780

**Published:** 2022-10-22

**Authors:** Asad Murtaza, Haroon Afzal, Thu-Dung Doan, Guan-Ming Ke, Li-Ting Cheng

**Affiliations:** 1International Program in Animal Vaccine Technology, International College, National Pingtung University of Science and Technology, Pingtung 91201, Taiwan; 2Centre for Immunology & Infection, Hong Kong Science and Technology Park, Hong Kong, China; 3General Research Service Center, National Pingtung University of Science and Technology, Pingtung 91201, Taiwan; 4Graduate Institute of Animal Vaccine Technology, College of Veterinary Medicine, National Pingtung University of Science and Technology, Pingtung 91201, Taiwan

**Keywords:** infectious bursal disease virus, immunosuppression, flagellin, adjuvant, N-terminus, chimeric protein

## Abstract

Flagellin activates the immune system through Toll-like receptor 5 (TLR5) and can work as an adjuvant for subunit vaccines. In this study, we tested the adjuvancy of two different N-terminal fragments of flagellin, (1) FliC_99_, residues 1–99, and (2) FliC_176_, residues 1–176, to incorporate larger areas of the hotspot region for potentially higher levels of TLR5 activation and immune response. A truncated version of the VP2 protein (name *t*VP2, residues 199–356) of the Infectious bursal disease virus (IBDV) was genetically linked to the flagellin constructs, and the immune response was evaluated in chickens. Results showed that both chimeric antigen–adjuvant constructs increased humoral (total IgG titers), cellular and cytokine immune response (IL-4, IFN-γ). The resulting antibody also successfully neutralized IBDV. We conclude that the N-terminus of flagellin can act as an immune activator to enhance vaccine efficacy.

## 1. Introduction

The flagellum is a whip-like organelle responsible for locomotion in bacteria [1]. Flagellin, the main protein component of the flagellum, is one of the pathogen-associated molecular patterns (PAMPs) that are recognized by the host immune system through receptors such as TLR5 [2] and NOD-like receptor protein four inflammasome receptor NAIP5/6 [3]. TLR5 detects the flagellin in the host and activates the MyD88-dependent signaling pathway, producing various proinflammatory cytokines and chemokines of the innate immune system. Flagellin has been used as an adjuvant for a number of antigens (bacterial, viral, and parasitic) with equal success [4,5]. 

Flagellin of *Salmonella enterica* serovar Typhimurium (*S. Typhimurium*) contains four domains, D0, D1, D2 and D3. D0 and D1, highly conserved and buried in the flagellum core filament, are responsible for innate immune activation through TLR5. On the other hand, D2 and D3, hypervariable and exposed on the surface of the filament, are accountable for antigenicity as well as undesirable toxicity [6,7,8]. 

Previous work has identified a crucial hotspot region (including conserved residues R89, L93 and E114) within D1 for binding to the leucine-rich repeat 9 (LRR9) region of TLR5 [6]. In our previous studies, we used an N-terminal portion of flagellin (residue 1–99, containing R89 and L93 of the hotspot region), and the results showed that it is sufficient for immune activation and vaccine efficacy enhancement [9,10,11]. In this study, we designed and tested two different N-terminal fragments of flagellin, (1) FliC_99_, residue 1–99, and (2) FliC_176_, residue 1–176, to incorporate more significant areas of the hotspot region, including E114, for potentially higher levels of TLR5 activation and immune response. 

To evaluate the above flagellin constructs as adjuvants in vivo, a truncated version of the VP2 protein (name *t*VP2, residue 199–356) of the Infectious bursal disease virus (IBDV) was employed as the vaccine antigen [12]. IBDV depletes B lymphocytes in the bursa of Fabricius, causing immunosuppression [13]. There are many vaccines available against IBD on the market. Still, due to the RNA genome of the virus, it is continuously evolving, making it challenging to control the disease with available vaccines [14]. In this study, flagellin constructs were genetically linked to *t*VP2 via a flexible glycine-serine (GS) linker. Chickens were immunized with antigen-adjuvant constructs to evaluate humoral and cellular immune responses. 

## 2. Materials and Methods

### 2.1. Bacteria Strain and Virus

*S. Typhimurium* (ATCC 14028) was obtained and cultured in Tryptic Soy Broth at 37 °C. The VP2 gene of Infectious bursal disease virus strain 3529/92 was obtained as a stable plasmid (VP2-pET32a, gift from Dr. Guanming Ke) and later confirmed by primers.

### 2.2. Epitope and Antigencity Prediction in the Antigen (IBDV)

To predict the B cell and antigenicity in the VP2 protein of the IBDV, BepiPred-2.0: Sequential B-Cell Epitope Predictor and Hydropath/Kyte & Doolittle methods were used, respectively. Using a Random Forest algorithm trained on epitopes and non-epitope amino acids with a threshold value set above 0.5 for all servers, the BepiPred-2.0 Sequential B-Cell Epitope Predictor server forecasts B-cell epitopes from a protein sequence. Briefly, an amino acid sequence was fed to IEDB tools for B-cell epitope prediction, and those which reside in a yellow color above the threshold (0.5 value) are the predicted epitopes, whereas the Y-axes depict residue scores and X-axes residue positions in the sequence. For predicting antigenicity, the sequence was fed to IEDB and Hydropath/Kyte & Doolittle methods using a threshold value of 0.5. The Y-axes represent the residue score and X-axes residue positions in the sequence. 

### 2.3. Cloning of the Desired Flagellin and IBDV Antigenic Segments and Expression of Antigen-Adjuvant Recombinant Proteins

To evaluate the adjuvancy of flagellin, three recombinant proteins were expressed as subunit vaccines: (1) FliC_99_-*t*VP2, the truncated N terminus of the flagellin (1–99 residues) fused with the truncated VP2 (199–356 residues) of the IBD virus, (2) FliC_176_-*t*VP2, the N terminus (1–176 residues) of the flagellin fused with the truncated VP2 (199–356 residues) of the IBD virus, and (3) *t*VP2, truncated VP2 only. First, *t*VP2 was cloned from full-length VP2 using the primers listed in Table 1 and ligated into the vector pET32a (Novagen, Darmstadt, Germany). For the cloning of the FliC_99_ and FLiC_176_, full-length flagellin or FliC clone (DNA of *S.* Typhimurium and inserted into pET32a) was used, and subcloning of the FliC_99_ (residues 1–99) and FliC_176_ (residues 1–176) was done using primers for FliC_99_ and FLiC_176_ (Table 1, Appendix A). To construct the FliC_99_-*t*VP2 and FliC_176_-*t*VP2, a chimeric polymerase reaction (PCR) was carried out using the *t*VP2, FliC_99,_ and FliC_176_ PCR products containing glycine-serine linker as a template. Primers for the chimeric reaction are listed in Table 1. pET32a was used as a vector to insert the final construct, and plasmid was sequenced for reconfirmation of the construct.

For the expression of the (1) FliC_99_-*t*VP2, (2) FliC_176_-*t*VP2 and (3) *t*VP2, the plasmids containing these constructs were transformed to *Escherichia coli* BL21 (DE3) (Yeastern Biotech, Taipei, Taiwan) according to the manufacturer’s instructions. 1-mM isopropyl-b-D-galactopyranoside (IPTG; Sigma, Darmstadt, Germany) was used to induce the protein expression at room temperature overnight. Cells were harvested, lysed in native lysis buffer (300-mM KCl, 50-mM KH2PO4 and 5-mM Imidazole) and sonicated. His-tag Bio-scale Mini Profinity IMAC cartridges (1 mL) (Bio-Rad, Hercules, CA, USA) were used to purify the recombinant protein from the insoluble fraction. Twelve percent sodium dodecyl sulfate-polyacrylamide gel electrophoresis (SDS-PAGE) analysis using BSA protein as standards was used to determine the expression levels of the recombinant proteins. Image Lab software predicted the concentration of the recombinant proteins. A Western blot assay was performed to confirm the identity of the recombinant proteins. In brief, after gel electrophoresis, proteins were transferred onto polyvinylidene difluoride (PVDF) membranes (Merck, Darmstadt, Germany). 6X-His Tag antibody solution (Gentex, Hsinchu, Taiwan) at 1:5000 dilution was used as the primary antibody, and rabbit anti-mouse antibody conjugated to HRP (Gentex, Taiwan) was used as the secondary antibody at 1:5000 dilution. Western Lightning PLUS (PerkinElmer, Waltham, MA, USA) was used for color development. PyMOL 2.5 (Schrodinger, New York, NY, USA) was used to predict the crystal structure of the FliC_176_, FliC_99_ and *t*VP2. Briefly, PDB ID (Protein Data Bank identifier) of flagellin Salmonella enterica serovar Typhimurium (*S. Typhimurium*) and Infectious Bursal Disease Virus (IBDV) were fetched on the PyMOL 2.5 and structure was drawn according to the amino acids sequence. 

### 2.4. Vaccine Preparation and Immunization of Chickens

Four vaccine formulations were prepared: (1) FliC_176_-*t*VP*2*, (2) FliC_99_-*t*VP2, (3) *t*VP2 and (4) PBS as the control. Each purified recombinant protein dose was 50 mg. Water-in-oil adjuvant Summit-P101 (Country Best Biotech, Taipei, Taiwan) was used for the formulation at the ratio of 2:1 (adjuvant: antigen) to make up the final injection volume of 0.2 mL per chicken. 

For immunization, 16 two-week-old Brown Leghorns chickens were obtained from a local farm and randomly assigned to four groups of three chickens for the four different vaccine formulations. Chickens were immunized subcutaneously twice, two weeks apart. Whole blood was collected from three chickens per vaccine group on days 0, 7, 14, 21 (Table 2) and 28 post vaccinations to analyze the immune response. All experimental animal protocols (NPUST-106-055) were approved by the Animal Care and Use Committee, National Pingtung University of Science and Technology (NPUST). The Ethical Rules and Laws of NPUST were followed during the experiments. 

### 2.5. Analysis of Antibody Immune Response

An indirect enzyme-linked immunosorbent assay (ELISA) was done to determine the antibody response in the chicken. Whole blood was allowed to coagulate and then centrifuged at 700× *g* for 5 min to separate the serum from erythrocytes. The ELISA plates were coated with 50-ng/well-purified *t*VP2 overnight at 4 °C. The next day, washing and blocking were done, and the primary antibody (serum) was used at 1:10,000 dilution. For the secondary antibody, horseradish peroxidase (HRP)-conjugated anti-chicken IgG (Sigma, Carlsbad, CA, USA) was used at a 1:5000 dilution. The Peroxidase Kit (KPL, Gaithersburg, MD, USA) was used for color development. In the end, the optical density of the plates was read at 450 nm on the Multiskan^TM^ FC microplate photometer (Thermo Fisher Scientific, Vantaa, Finland). 

### 2.6. Analysis of Cell-Mediated Immune Response

To check the T-cell proliferation activity, an MTT ((3-(4,5-dimethylthiazol-2-yl)-2,5-diphenyltetrazolium bromide) assay was done. Blood was collected from chickens (*n* = 3, two weeks-old Brown Leghorns) on days 2, 14 and 28 to separate the peripheral blood mononuclear cells (PBMCs) and stimulated with *t*VP2. Briefly, Blood was collected in BD Vacutainer™ EDTA Blood Collection Tubes (BD Biosciences, Franklin Lakes, NJ, USA), and an equal volume of PBS was added to the blood. The PBS–blood mixture was slowly added to another tube’s equal volume of Ficoll-Paque (Amersham Biosciences, Piscataway, NJ, USA). The mixture was centrifuged at 252× *g* for 40 min to collect the buffy coat layer containing PBMCs fraction. Cells were rinsed with PBS twice and resuspended in RPMI-1640 (Gibco Invitrogen, Carlsbad, CA, USA) supplemented with 5% fetal bovine serum (Gibco Invitrogen, Carlsbad, CA, USA) at 2 × 10^6^ cells/mL. For antigen stimulation, PBMCs (4 × 10^5^ cells/well) were added to 96-well plates incubated with 10 μg/mL of the purified recombinant *t*VP2 for 48 h at 37 °C, 5% CO_2_. Concanavalin A (Thermo Fisher Scientific, Ward Hill, MA, USA) at 5 µg/mL was used as the positive control for cell stimulation, with cell-only as the negative control and medium-only as the background. The cell proliferation was measured using EZcountTM MTT Cell Assay Kit (Himedia, India). The results were read at 550 nm on the MultiskanTM FC microplate photometer (Thermo Fisher Scientific, Vantaa, Finland). Stimulation index (SI) = (OD of treatment − OD of background)/(OD of the negative control − OD of background).

### 2.7. Th1 and Th2-Type Cytokine mRNA Expression Level

PBMCs were collected from the vaccinated group on Day 28, as described in Section 2.5. Briefly, the cells (2 × 10^6^ cells/well) were added to the 6-wells plate containing 10 μg/mL of the *t*VP2 for the stimulation and incubated for 2 h incubation at 37 °C, 5% CO_2_. A Miniprep system (Viogene, Taipei, Taiwan) was used for the extraction of total RNA, and a Reverse Transcriptase Kit (Applied Biosystems, Foster, CA, USA) was used to convert the RNA into complementary DNA (cDNA). Chicken cytokines primers listed in Table 3 were used for the Real-time PCR in the SmartCycler I (Cepheid, Sunnyvale, CA, USA) for T_H_1 (IFN-γ) and T_H_2-type cytokines (IL-4) and the housekeeping gene glyceraldehyde-3-phosphate dehydrogenase (GAPDH). GAPDH was used as the reference gene to normalize the cytokine expression levels, and changes were expressed as n-fold higher or lower in levels relative to the PBS control.

### 2.8. IBD Virus Neutralization Assay

Serum samples from Day 28 were used for the neutralizing antibody assay. The titers of the virus-neutralizing antibodies were examined using IBD virus and Vero cell lines. Briefly, the serum was heat-inactivated, and a 2-fold dilution of the serum was done on a 96-well plate. Then, serum samples of each dilution were mixed with an equal volume of the virus (100 TCID_50_) and incubated for 1 h at 37 °C. Vero cell lines were added and again incubated for 2 h at 37 °C_._ After that, the supernatant was removed, and the virus was allowed to grow on the cell lines for 5 to 7 days. Cells were monitored for cytopathic effects, and the virus-neutralizing antibody titer of the serum was calculated according to the method of Reed and Munch (1938).

### 2.9. Statistical Analysis

For statistical analysis, the IBM SPSS Statistics software was used. One-way analysis of variance (ANOVA) was used to compare data on antibody response, lymphocyte proliferation assay and cytokine mRNA levels. For the comparison of neutralization assay data, the *t*-test was used. The significance level (*p*) was set at 0.05 for all the experiments. All the data are expressed as mean ± standard error of the mean (SEM). 

## 3. Results

### 3.1. Antigen-Adjuvant Chimeric Proteins Were Formulated as Subunit Vaccines

B-cell epitope prediction and hydrophobicity plot showed that the region 199–356 (Figure 1) of IBDV VP2 contains several epitopes and can be highly antigenic. Therefore, this region was selected as the antigen for vaccine construction.

Two separate chimeric proteins were constructed to evaluate the adjuvant effect of flagellin N terminus 1–99 and 1–176: the N flagellin 1–99 linked to *t*VP2 (199–356) of IBD virus and N flagellin 1–176 also linked to *t*VP2. Protein expression was confirmed by SDS-PAGE (Figure 2C) and Western blot (Figure 2D) with a size of 48kDa (FliC_99_-*t*VP2). The size of the FliC_176_-*t*VP2 was 55kDa (Figure 2A,B). Full-length VP2 and *t*VP2 were also expressed, and the size was confirmed by running SDS-PAGE (Figure 2E) and Western blot (Figure 2F). 20-kDa was an additional size protein carried with it when we cloned in pET32a, because the pET32a vector inserts a 20-kDa Trx-His-S-enterokinase tag. The concentration value was up to 140 µg/mL for FliC_99_-*t*VP2 and 150 µg/mL for the FliC_176_-*t*VP2 variably in different purification attempts. Four vaccine formulations were prepared for the immunization of chickens: (1) FliC_99_-*t*VP2, (2) FliC_176_-*t*VP2, (3) *t*VP2 and (4) PBS as the negative control.

Molecular structure of FliC_176_, FliC_99_ and *t*VP2 were predicted using PyMOL 2.5 (Figure 3A–C). 

### 3.2. Both FliC_176_ and FliC_99_ Resulted in an Increase in Antibody Levels

Indirect ELISA was used to determine the levels of antibody response, with *t*VP2 as the coating antigen. On Day 21 and 28, the levels of the antibodies went up in the FliC_176_-*t*VP2 and FliC_99_-*t*VP2 groups as compared to the *t*VP2-only group (Figure 4), indicating that the flagellin enhanced antibody production.

### 3.3. Both FliC_176_ and FliC_99_ Increased the Proliferation of the Lymphocytes

To measure lymphocytes’ metabolic activity, the MTT assay was performed using PBMCs. Cells were stimulated with *t*VP2, and the proliferation activity was measured. After the 2nd dose of the vaccine, on the 14th and 28th days, the proliferation activity of both groups, FliC_176_-*t*VP2 and FliC_99_-*t*VP2, rose considerably compared to the *t*VP2 and PBS groups (Figure 5). This indicates that the flagellin was essential in increasing the number of lymphocytes in the vaccinated chickens. 

### 3.4. Both FliC_176_ and FliC_99_ Raised the Gene Expression of the Cytokines

Two weeks after the 2nd dose, on the 28th day, the PBMCs were collected and stimulated with tVP2, and cytokine mRNA levels were quantified. Th1 (IFN-γ) levels and Th2 (IL-4) cytokines increased significantly in the vaccinated chickens (Figure 6).

### 3.5. Neutralization Assay

On day 28, after vaccinations, the serum samples were used for the neutralization assay with 100 TCID50 of the Infectious Bursal Disease Virus. The serum sample of the FliC_176_-*t*VP2 group neutralized the virus up to the 5th dilution of the log2, while on the other hand, serum of FliC_99_-*t*VP2 neutralized the virus up to the 4th dilution of the log2. In the *t*VP2 serum, only the 1st dilution of the log2 neutralized the virus, whereas neutralization was not noticed in the PBS-only group (Figure 7 and Appendix A).

## 4. Discussion

This study demonstrated that N-terminal fragments of flagellin encompassing the hotspot region can enhance immune response when linked to an antigen. FliC_176_ and FliC_99_ both boosted antigen-specific antibody levels, virus-neutralizing activity, cellular immunity and cytokine gene expression. 

Our experimental data revealed that there is not a statistically significant difference between FliC_176_ and FliC_99_ in terms of immune modulation (antibody titers, T-cell stimulation and cytokine expression). We hypothesize that the hotspot is the main contributor to the TLR5 activation. Liu et al., showed that residues 85 to 111 of flagellin encompassing the hotspot region resulted in significantly higher titers of neutralizing antibodies and cytokines against PCV2 than antigens alone [15]. A similar trend of immune activation was measured in the anti-tumor vaccine (B16-OVA melanoma, P815 tumor model) [16,17]. Therefore, any length of flagellin encompassing the hot-spot region of flagellin may be sufficient for adjuvancy. Both of our truncated designs of flagellin-VP2 elicit statistically insignificant antibody and cytokine titers. However, neutralizing antibody titers did show significant difference among the vaccine groups. 

Previous work using flagellin as a therapeutic agent pointed out that the development of anti-flagellin antibody upon repeated exposure can hamper effectiveness. A case in point is Entolimod, an anti-radiation drug developed from flagellin D0, D1. GP532 (modified flagellin) was developed as an anti-radiation drug with minimal anti-flagellin antibodies to circumvent this issue. Deimmunization was accomplished after deleting B-cell, T-cell and inflammasome activation epitopes. In light of this, future truncated flagellin designs should be screened for anti-flagellin antibodies [18]. 

Truncation of flagellin can be carried out in many other ways, and the best possible truncation design could be narrowed down for therapeutic use. The N-terminal domain of D0 was removed in the GP532 design without impacting TLR5 binding and activation. Hot-spot is pivotal to flagellin-TLR5 adjuvancy. Thus, the binding domain of flagellin (D1) could be linked to either the N or the C-terminus of D0. In short, various truncations can be performed in D0, D1 with respect to the goal of the experiment by retaining the hot-spot region in truncation. In addition to truncation designs, adjuvancy of naturally conserved (D0, D1) flagellin from *Bacillus cereus* and *Bacillus subtilis* should be considered for future studies. These short flagellins offer adjuvancy comparable to FliC. Moreover, reduced proinflammatory properties were observed in mice compared to flagellin from *S. Typhimurium* [19]. Further studies are required to determine the de novo antigenicity of these flagellins.

## 5. Conclusions

This study finds that the N-terminus of flagellin, both FliC_99_ and FliC_176_, can act as an immune activator to enhance vaccine efficacy for an IBDV VP2 subunit vaccine.

## Figures and Tables

**Figure 1 vaccines-10-01780-f001:**
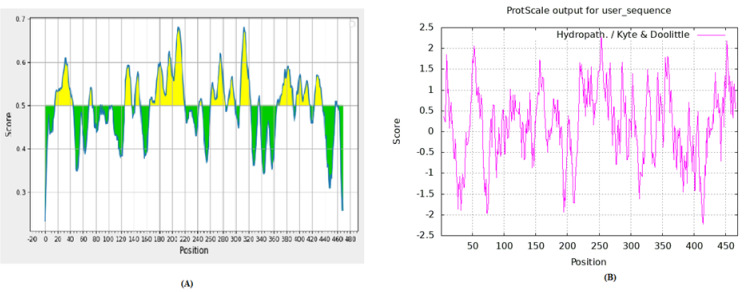
(**A**) The figure represents the B−cell epitope prediction in the hypervariable region of VP2 by BepiPred−2.0: Sequential B−cell Epitope Predictor. Amino acids sequence was fed to IEDB tools for B−cell epitope prediction. The yellow color represents the epitopes above the threshold while green ones are below the threshold value. Epitopes were selected from the hypervariable region having values above 0.5 thresholds. (**B**) The figure represents the hydrophobicity index of full−length VP2 by Hydropath/Kyte & Doolittle method. Amino acids in the hypervariable region represent values above the threshold of 0.5.

**Figure 2 vaccines-10-01780-f002:**
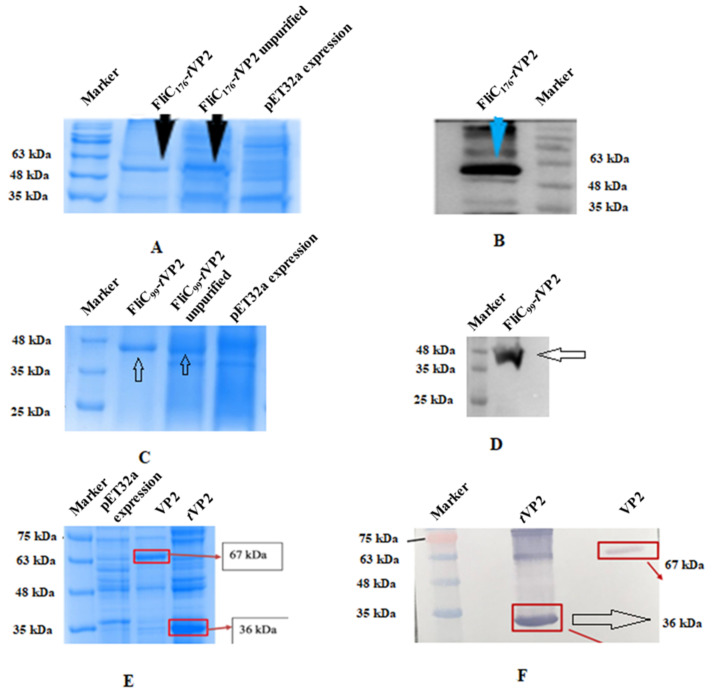
(**A**) SDS-PAGE of FliC_176_-*t*VP2. (**B**) Western blot analysis of FliC_176_-*t*VP2. (**C**) SDS-PAGE of FliC_99_-*t*VP2. (**D**) Western blot analysis of FliC_99_-*t*VP2. (**E**) SDS-PAGE of VP2; and *t*VP2 (**F**) Western blot analysis of VP2 and *t*VP2.

**Figure 3 vaccines-10-01780-f003:**
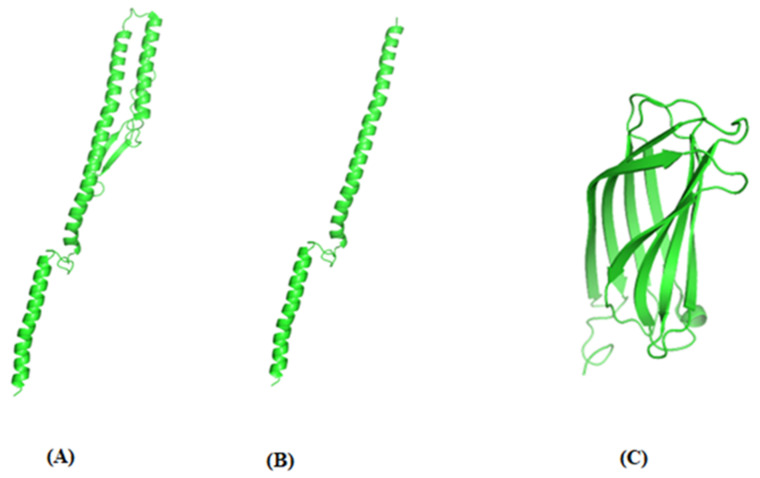
Predicted structure of (**A**) FliC_176_ and (**B**) FliC_99_ based on PDB ID: 1UCU. (**C**) Predicted structure of *t*VP2 199–356 based on PDB ID: 2DF7 using PyMOL2.5.

**Figure 4 vaccines-10-01780-f004:**
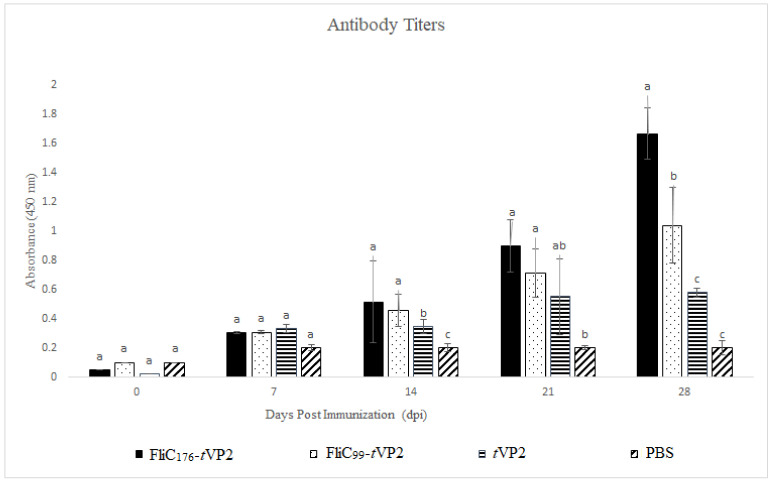
Three chickens from each group (*n* = 3) were immunized with the FliC_176_-*t*VP2, FliC_99_-*t*VP2 *t*VP2 and PBS. Sera were analyzed by indirect ELISA using *t*VP2 as coating antigen (10 µg). The bar chart shows the antigen-specific antibodies of the immunized chicken. Data are presented as mean ± SEM. Different letters indicate significant differences (*p* < 0.05) between treatment groups at the same time point.

**Figure 5 vaccines-10-01780-f005:**
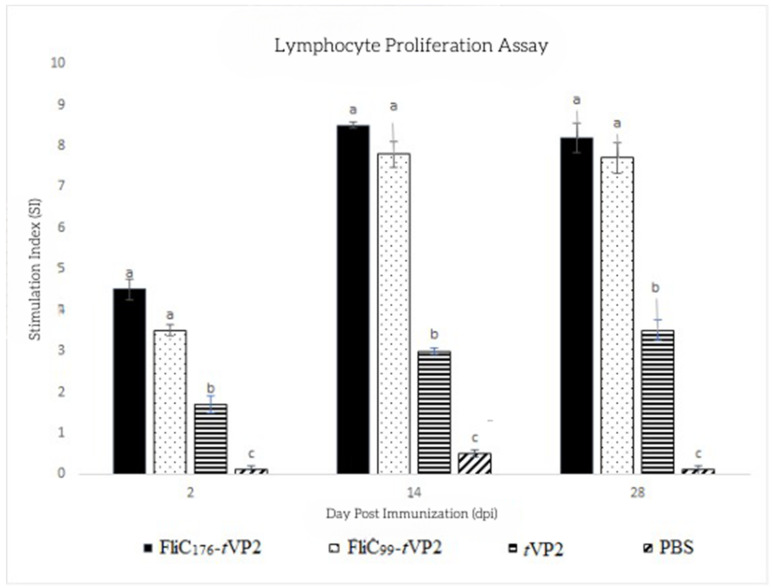
Each group’s chickens (*n* = 3) were immunized with the FlIC176-tVP2, FliC99-tVP2, tVP2 and PBS. PMBCs were collected, and after stimulation with tVP2, lymphocyte proliferation was analyzed using an MTT assay. Data are presented as mean ± SEM. Different letters indicate significant differences (*p* < 0.05) between treatment groups at the same time point.

**Figure 6 vaccines-10-01780-f006:**
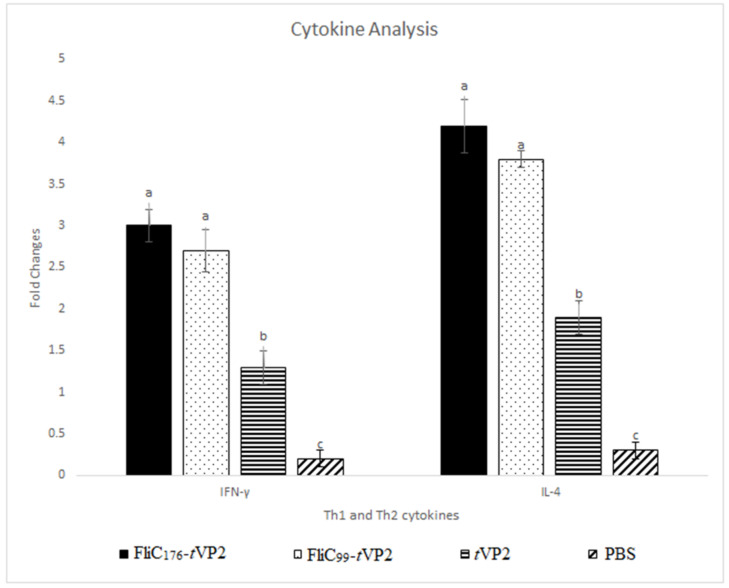
Peripheral blood mononuclear cells (PBMCs) were taken from chickens (number of chickens in each group = 3) and treated with 10 μg/mL of the tVP2. Expression levels of the cytokines, such as IFN-γ and IL-4, were determined. Data are presented as mean ± SEM. Different letters indicate significant differences (*p* < 0.05) between treatment groups at the same time point.

**Figure 7 vaccines-10-01780-f007:**
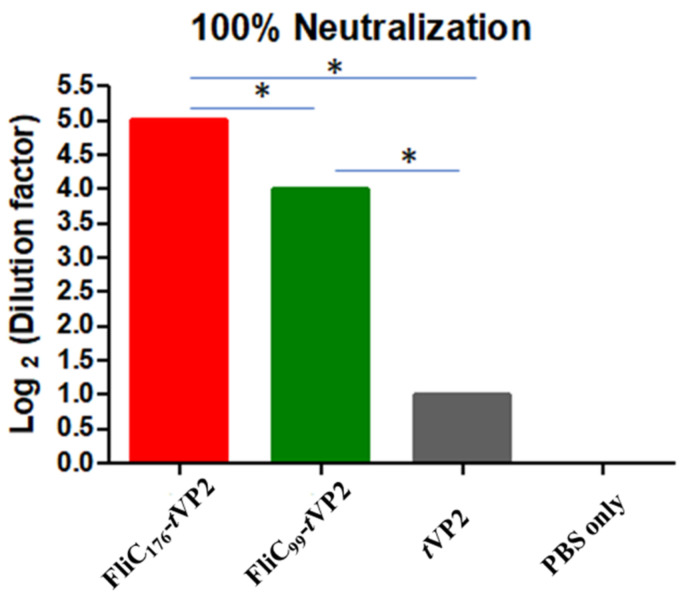
Serum samples were taken from the chicken (number of chickens in each group = 3), and 100 TCID50 of the virus was used to check the neutralization activity of the antibody. Data are presented as mean ± SEM, and the *t*-test showed a significant difference (*p* < 0.05) between different groups. *. Indicate the significant differences (*p* < 0.05) between treatment groups at the same time point.

**Table 1 vaccines-10-01780-t001:** Primers for gene cloning and recombinant protein construction.

Target	Sequence (5′-3′)	RE	Gene-	DNA
Gene		Site	Length (bp)	Templates
FliC	F ^1^ *ggatcc*atgaaaaagacaatcgtagc	BamHI	1500	*ATCC 14028*
	R *gtcgac*ttagaagtgtacgcgtaaac	SalI		
*t*VP2	F ^2^ **ggcgggggcggcagc**accata actgcagccgatg	-	489	*ATCC 5848*
	R *ctcgag*cgtaacgacagaccctgt	Xhol		
FliC_176_	F *ggatcc*aacccgctgcagaaaattg	BamHI	543	*FliC*
	R **gctgccgcccccgcc**acgcag taaagagaggac	-		
FliC_176_-*t*VP2	F *ggatcc*aacccgctgcagaaaattg	BamHI	1017	*t*VP2 and FliC_176_
	R *ctcgag*cgtaacgacagaccctgt	Xhol		
FliC_99_	F *gaattc*atggcacaagtcattaatacaaac	EcoRI	312	*FliC*
	R **gctgccgcccccgcc**agactgaaccgccagttc	-		
FliC_99_-*t*VP2	F *gaattc*atggcacaagtcattaatacaaac	EcoRI	786	*t*VP2 and FliC_99_
	R *ctcgag*cgtaacgacagaccctgt	Xhol		

^1^ Italics in the primers represent restriction enzyme (RE) sites. ^2^ Bold fonts in the primers represent glycine–serine linkers.

**Table 2 vaccines-10-01780-t002:** Design of chicken immunization.

Group	1st Dose	2nd Dose	Adjuvant	Immunization Dose	Route
FliC_176_-*t*VP2	0 Day	14 Days	Summit-P101	50 μg	S/C
FliC_99_-*t*VP2	0 Day	14 Days	Summit-P101	50 μg	S/C
*t*VP2	0 Day	14 Days	Summit-P101	50 μg	S/C
PBS	0 Day	14 Days	Summit-P101	50 μg	S/C

**Table 3 vaccines-10-01780-t003:** Primers for cytokine genes.

Target	Sequence (5′-3′)	Length	Annealing Temp.	GenBank
Gene		(bp)	(°C)	
	R cttcca agggatcttcattt			
IFN-γ	F gacggtggacctattatt	255	50	HQ739082
	R ggctttgcgctggattc			
IL-4	F tgtgcccacgctgtgcttaca	193	61	AJ621249.1
	R cttgtggcagtgctggctctcc			
GAPDH	F tgctgcccagaacatcatcc	142	55	NM_204305
	R acggcaggtcaggtcaacaa			

GAPDH: glyceraldehyde 3 phosphate dehydrogenase; IL: interleukin; IFN-γ: interferon gamma.

## Data Availability

Not applicable.

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
