# Peer review of "Flagellin Improves the Immune Response of an Infectious Bursal Disease Virus (IBDV) Subunit Vaccine"

_vaccines, 2022, doi:10.3390/vaccines10111780_

Round 1

Reviewer 1 Report

In the present manuscript, Murtaza et al. have analyzed the effect of Flagellin N-terminal fragments as an adjuvant for subunit vaccination with a truncated version of VP2 from Infectious bursal disease virus. They analyze two different fragments, FliC99 - used successfully before- and FliC176, a longer version. They saw differences for the two fragments at least in the neutralization test, but no prominent advantage of using the longer FliC176 version instead of FliC99.

The introduction and methods are clear and well written, but the results text  has some mistakes concerning the link between the text referring to the correct figure.

The novelty and significance of their findings is rather low as the adjuvant effect of the Flagellin fragment has been shown before in several publications.  

Detailed comments

2.6 page 4 what does MTT stand for?

3.1 page 6 wrong Figure references; figure 2 (A,B,E,F) is named figured 1

3.2 Page 7 :

- Figure 2 in the text should be figure 4

- What about figure 3, it is inserted in the manuscript but not referred to in the manuscript text

- Figure 4: it is not plausible what the different letters mean in terms of significance here, it doesn´t seem to work out at least for some of the data. For example, for 0dpi it is not plausible that FliC176-tVP2 and FliC99-tVP2 are not significantly different but FliC99-tVP2 and PBS are. The authors should check on that.

- presence = present

4. discussion

The authors state here that against their expectations there was no difference in neutralization between the vaccine groups but figure 4 shows significant differences, how can this be explained?

Reviewer 2 Report

The authors did a very good job in identifying a way to increase antigenecity for a vaccine against IBDV. Although the idea of using N-terminal fragments of flagellin is not new it is however a new construct used for IBDV.

Here are some revisions expected from the authors of the paper:

1) The eventual goal of a vaccine is to improve survival rate the authors did not shown any survival rate of chicken after viral challenge.

2) Percentages of CD4+ and CD8+ T cells in the PBMCs of immunized chickens were not shown

Minor comments:

1) the source of VP2 plasmid line 65 was not mentioned

2) The details of methods for 2.2 (materials and Methods) are not mentioned.

Round 2

Reviewer 2 Report

Unfortunately the authors have not addressed issues regrading experiments needed to complete this manuscript. However, if the authors complete the survival rate of chicken after viral challenge, the manuscript can be considered for publication. 

Round 3

Reviewer 2 Report

The authors have made their case which has been accepted and we can proceed with the publication. Congratulations!